# Polio Epidemiology: Strategies and Challenges for Polio Eradication Post the COVID-19 Pandemic

**DOI:** 10.3390/vaccines12121323

**Published:** 2024-11-26

**Authors:** Lucia F. Bricks, Denis Macina, Juan C. Vargas-Zambrano

**Affiliations:** 1Sanofi Vaccines Medical, Av Nações Unidas, São Paulo 14401, Brazil; 2Sanofi Vaccines Medical, 14 Espace Henri Vallee, 69007 Lyon, France; denis.macina@sanofi.com (D.M.); juan.vargas@sanofi.com (J.C.V.-Z.)

**Keywords:** poliomyelitis, epidemiology, vaccine coverage, surveillance, polio eradication

## Abstract

The Global Polio Eradication Initiative (GPEI), launched in 1988, has successfully reduced wild poliovirus (WPV) cases by over 99.9%, with WPV type 2 and WPV3 declared eradicated in 2015 and 2019, respectively. However, as of 2024, WPV1 remains endemic in Afghanistan and Pakistan. Since 2000, outbreaks of circulating virus derived of polio vaccines (cVDPVs) have emerged in multiple regions, primary driven by low vaccine coverage rates (VCRs). The COVID-19 pandemic disrupted routine immunization, resulting in millions of unvaccinated children, and leaving many countries vulnerable to both WPV1 and cVDPVs outbreaks. This paper reviews the epidemiological landscape of poliomyelitis post the COVID-19 pandemic, and the strategies and challenges to achieve the global polio eradication.

## 1. Introduction

Poliomyelitis presenting as acute flaccid paralysis (AFP) can be caused by three WPV types: 1, 2, and 3. Two of these WPVs were globally eradicated: WPV2 since 2012, and WPV3 since 2019 [1]. In 2024, only WPV1 persists in two endemic countries, but outbreaks caused by circulating vaccine-derived poliovirus (cVDPVs) have been reported since 2000, the year initially proposed for global polio eradication [1,2,3,4,5,6,7,8,9].

The success in reducing by more than 99.9% the number of AFP cases caused by WPV involved massive vaccination campaigns with the trivalent oral polio vaccine (tOPV), which is composed of live attenuated polio strains—Sabin 1, 2, and 3 [1,3,8]. In 1994, the American region was declared free of WPV by the WHO [1], but in 2000 the first outbreak caused by cVDPV1 was detected in this region [1,8,10].

Before 2000, few countries had adopted the inactivated polio vaccine (IPV) in their routine immunization (RI) schedules. The majority were European countries [8,9,11], where rare cases of AFP caused by polio were recorded, mainly in unvaccinated or inadequately vaccinated individuals, and WPV1 was the most frequent strain [8,9]. Most countries adopted tOPV on the basis of its ease of administration, its low cost, its robust immunogenicity, and its capacity to induce intestinal mucosal immunity (IMI), which helps reduce poliovirus shedding in breakthrough infections [9]. Sabin vaccine is considered one of the safer vaccines, but Sabin live virus strains are genetically unstable, and in rare occasions they can mutate and reacquire neurovirulence. As a result, OPV can cause vaccine-associated paralytic poliomyelitis (VAPP) in recipients or contacts of recipients. The risk is higher after the first dose, in communities with low vaccine coverage rates (VCRs), and among immunocompromised individuals. These revertant strains named vaccine-derived polioviruses (VDPVs) can cause paralysis, with indistinguishable clinical signs and severity from those caused by WPV [8].

VDPVs are classified in three categories: cVDPVs (circulating), when there is evidence of transmission (more than one case), iVPVD, when the infections are confirmed in immunocompromised individuals, and aVDPV (ambiguous) that can represent the initial isolates from cVDPV outbreaks, samples isolated from individuals without documented immunodeficiencies or that are environmental samples, with no evidence of community circulation [8]. Low VCRs are the most important factor associated with the emergence and circulation of these new pathogenic strains [1,3,8,9].

At the onset of the 21st century, Canada and the USA replaced tOPV by IPV, considering that the number of VAPP cases outnumbered those caused by WPV, and these countries changed tOPV to IPV. The IPV vaccine is highly immunogenic, safer, and more effective than OPV to prevent polio, but it induces low IMI. After 2000, as outbreaks caused by cVDPVs emerged in all WHO regions [3,8,9,10], high-income countries (HICs), where high VCRs were achieved with three doses of tOPV [6,7], progressively included IPV in their national immunization programs (NIPs) [8,9,11].

Initially, IPV was included in NIPs concomitantly with DPT-based whole-cell or acellular pertussis vaccine (DTPw or DTPa). As the acellular pertussis vaccine quickly proved amendable to be combined with IPV, DTPa combination vaccines including IPV and other antigens were developed and introduced into RI. HICs adopted acellular pertussis vaccines in combination with IPV earlier, while low-income countries (LICs) and middle-income countries (MICs) continued using OPV and combination vaccines with DTPw [8,9,11]. Nowadays, HICs use pentavalent or hexavalent combination vaccines formulated with IPV and aP, while most LICs and MICs still rely on pentavalent vaccines containing whole cell pertussis vaccines and the separate administration of at least one dose of IPV and more than two OPV doses [11,12].

The GPEI strategy recommended for global polio eradication focused on tOPV until 2016. The amazing success in eradicating WPV2 and WPV3 in four of the six WHO regions—the Americas (1994), the Western Pacific (2000), Europe (2002), and Southeast Asia (2014), associated with the fact that OPV induces IMI and the rare occurrence of VAPP cases before 2015, were the main reasons invoked to maintain the use of tOPV in RI. Between 2004 and 2009, monovalent oral polio vaccines (mOPV) were also used during specific outbreaks [1,3,8,13,14].

Despite this success, even in areas declared polio-free, many AFP cases were associated with the continued use of OPV [2]. This was especially the case in areas with low VCR, where cVDPV2 caused the majority of outbreaks [2,3,8,9,10,11].

Polioviruses are highly infectious [8] and the WHO recommends a minimum of 90% of VCR with three doses of OPV in the first year of life to achieve global polio eradication [1]. In 2018, the global VCR was 84%, but with large variations between different countries and regions of the same country [6,7]. In addition, the COVID-19 pandemic caused a reduction in the global VCR for all vaccines, resulting in millions of children remaining unprotected against polio and other serious diseases [3,6].

Since 2017, the number of AFP cases caused by VDPVs has surpassed those caused by WPV1 [2,3,4,5] (Figure 1).

Poliovirus transmission, whether WPV and cVDPV, has been declared by the WHO as a Public Health Emergency of International Concern (PHEIC) since 2014, and is still a PHEIC in 2024 [3].

The objective of this article is to review the polio epidemiology after the COVID-19 pandemic, the strategies and challenges to achieve global polio eradication, and the plans for the post-eradication era.

## 2. Polio Epidemiology After the COVID-19 Pandemic

Since 2019, WPV1 is the only WPV still in circulation, but the number of AFP cases and outbreaks caused by VDPVs is a cause of concern [4]. The minimum number of AFP cases caused by WPV1 before the onset of the COVID-19 pandemic was 22, in 2017. In 2019 and 2020, an increase in AFP cases caused by WPV1 was observed, with 176 and 140 cases, respectively. Most of these cases were notified in Afghanistan and Pakistan, but nine children and adolescents outside these endemic countries developed AFP caused by WPV1 in 2021 and 2022: one in Malawi (November 2021) and eight in Mozambique (2022) [5]. Extensive national and subnational campaigns of immunization were conducted in these countries and in neighboring countries (Tanzania, Zambia, and Zimbabwe) to avoid WPV1 spreading in the region. No new AFP or detection of WPV1 through environmental surveillance (ES) was registered in these countries after 2022, and they were not considered as endemic countries [12].

In 2024, the WHO considers only Pakistan and Afghanistan as endemic countries, i.e., countries where the transmission of indigenous wild poliovirus never stopped. Thanks to the work of the WHO and many partners (UNICEF, Rotary International, the US Centers for Disease Control and Prevention, the Bill and Melinda Gates Foundation, and Gavi, the Vaccine Alliance) to increase VCR in endemic and recent outbreak countries, including numerous and extensive campaigns of immunization, the number of APF cases caused by WPV1 was reduced to only six, in 2021. Yet this is no reason to let the guard down, as in 2024, this number is more than five times higher, with 62 AFP cases caused by WPV1 confirmed in Afghanistan (23) and Pakistan (39) as of 22 October 2024 compared with 12 AFP cases in 2023 (full year) [4].

Despite the success of maintaining a geographically limited circulation of WPV, the number of countries that previously stopped indigenous WPV but are experiencing either the importation of WPV or VDPVs from other countries or the local emergence of cVDPVs has grown since 2016.

In 2016, there were 37 AFP cases caused by WPV1 and five caused by cVDPVs. In 2017, the number of AFP cases caused by cVDPV2 (96) surpassed the number of AFP cases caused by WPV1 (22), reaching a peak in 2020 (1082), affecting 24 countries (Table 1).

There was a reduction in the number of AFP cases caused by cVDPV2 over the last 4 years, but new countries have been infected, reaching a total of 51 countries with detection of cVDPV2. In 2024, cVDPV2 strains were detected only through environmental surveillance in 13 countries (Figure 2) [2,3,4,5,13,14].

In 2023, the WHO reported 12 AFP cases caused by WPV1 and 527 AFP cases caused by cVDPVs. From 1 January to 22 October 2024, 62 AFP cases caused by WPV1 were confirmed and cVDPV2 persists as the predominant strain detected through AFP surveillance and having caused paralysis in 182 individuals from 16 countries, including one case in Gaza confirmed on 25 July 2024 [3] (Table 1).

In the last four years, outbreaks caused by cVDPV1 were registered in six countries: Congo, the Democratic Republic of the Congo, Madagascar, Malawi, and Mozambique. Despite the substantial reduction in these numbers, as recently as 25 September 2024, cVDPV1 was still circulating in the Democratic Republic of the Congo (five AFP cases) and Mozambique (one APF case); additionally, one AFP case caused by cVDPV3 was confirmed in Israel in 2022 [2,5].

## 3. Strategies Recommended by the WHO for Polio Eradication

The GPEI strategies to achieve polio eradication changed many times over the years. From 1988 to 2005, the WHO recommended the exclusive use of tOPV in routine and immunization campaigns. Monovalent OPV vaccines (mOPV) became the vaccines of choice for supplemental immunization campaigns between 2005 and 2009 in polio-endemic or polio-epidemic countries. Indeed, the single-strain vaccine can help avoid the issue of the immunological interference between the three Sabin strains, especially with Sabin-2, the most immunogenic [8]. The massive use of mOPV1 (mOPV) resulted in the virtual elimination of WPV1 in India. Bivalent OPV containing Sabin strains 1 and 3 (bOPV1,3) has superior immunogenicity against polio 1 and 3 as compared with tOPV, without loss of immunogenicity as compared with mOPV1 and mOPV3, and was also largely used after WPV2 global eradication [8].

In 2012, with the dramatic reduction in WPV and eradication of WPV2, the WHO started to recommend the introduction of one dose of IPV to prepare for the post-eradication era. In that year, an IPV based on Sabin strains (sIPV) started to be used in Asia (Japan in 2012 and China in 2015) [9]. The WHO considers that these vaccines are as immunogenic and safe as IPV manufactured from WPV. HICs adopted a full IPV schedule before 2015, preferentially in combination with other vaccines, but most of the MIC and LIC persisted using tOPV and mOPVs until 2016. That year, with the declaration of the eradication of WPV2, the WHO recommended the substitution of tOPV by bOPV1,3 and one IPV dose [8,9,13,14]. The limitation in the supply of IPV, and a delay in the introduction of this vaccine in the poorest countries, resulted in a substantial gap of protection against polio 2, the most likely cause of the emergence of several cVDPV2 outbreaks in more than 40 countries. To overcome the short supply of IPV, the WHO recommended the use of one-fifth of an IPV dose (fractional IPV—fIPV), a strategy that was adopted by a limited number of LICs and MICs [8,9,11,13,14].

The last Polio Eradication Strategy 2022–2026 has two main goals: (1) permanently interrupting WPV1 transmission in endemic countries and (2) stopping cVDPV transmission and preventing outbreaks in non-endemic countries, both by the end of 2023, with the aim of reaching eradication by 2026 [1,3,13,14]. However, in 2024, these two goals were not achieved [1,2,3,4,5].

One dose of IPV or two fIPV are not sufficient to protect against polio 2 or guarantee durable protection [9].

The challenges encountered in the substitution of tOPV for bOPV1,3 resulted in a higher number of cVDPV2 outbreaks and countries affected by them [1,3,13,14]. The attempt to control cVDPV2 outbreaks with large immunization campaigns with mOPV2 was not successful. Between 1 January 2016 and 22 October 2024, 518 AFP cases caused by WPV1 were confirmed and 3983 AFP cases were caused by cVDPVs 1, 2, or 3, especially cVDPV2 [2]. The delay in the implementation of immunization campaigns and the incapacity to achieve high VCRs contributed to the emergence of new cVDPV2 cases, that spread to new areas and neighboring countries. The low VCR for bOPV1,3 in RI was responsible for the emergence of cVDPV1 outbreaks and one AFP case caused by cVDPV3 in the last 4 years. The absent or weak environmental surveillance in many countries also contributed to the late detection of VDPV emergences and late response to control its dissemination [13,14].

In 2020, the WHO recommended the introduction of a second IPV dose for all countries, considering that there was a sufficient supply of IPV, fIPV and sIPV [1,3].

In 2021, one novel OPV vaccine (nOPV2), formulated with Sabin 2 strains genetically attenuated to reduce the risks to revert to the neurovirulence observed with mOPV2, was introduced under an emergency list [15,16,17,18]. More than one billion doses of nOPV2 were used in 35 epidemic countries, including almost 500 million doses in Nigeria [15], but new countries have been infected by cVDPV between 2021 and 2024 (eight in 2023 and two in 2024) [5]. On 16 March 2024, the WHO informed that the GPEI received notification of the detection of cVDPV2 in Burundi and the Democratic Republic of the Congo (DRC) linked with the nOPV2. The viruses were isolated from the stool samples of seven children with acute flaccid paralysis—six in the DRC and one in Burundi, and from five environmental samples collected in Burundi [16]. Between January 2023 and June 2024, a total of 74 cVDPV outbreaks were detected in 39 countries, with 672 confirmed AFP cases identified in 27 of the 39 countries. Most outbreaks (70) were caused by cVDPV2, and 29 (comprising 19 VDPV2 emergencies) were linked to nOPV2 use in 19 countries [17].

On 19 July 2024, the WHO informed that a variant type 2 of poliovirus was isolated from sewage samples in Gaza [18], and one AFP case caused by cVDPV2 was confirmed in Gaza on 25 July 2024 [5]. On 13 August 2024, the WHO reported that a total of 16 new nOPV2-derived cVDPV2 emergences had been detected in the African Region, which now exclusively uses nOPV2 for outbreak response, and one had been identified in Egypt, in the Eastern Mediterranean Region. The AFP case caused by cVDPV2 confirmed in Gaza is genetically related to the lineage identified in Egypt [19].

## 4. Discussion

Many factors can be associated with the delay in global polio eradication, and for this paper we will focus our discussion on four: (1) vaccine coverage rates; (2) polio surveillance; (3) the benefits and risks associated with live and inactivated vaccines; (4) plannings for the final polio eradication and post-global eradication era.

### 4.1. Vaccine Coverage Rates

The best way to reduce polio infections is to maintain strong population immunity levels through high VCRs. This has proven difficult, especially in the poorest countries and conflict areas, since before 2019, and more so during the COVID-19 pandemic, when lockdowns and the reduction in basic health services reduced even more the VCRs in most countries. Even in high-income countries, vaccine hesitancy and loss of confidence in immunization increased because of fake news in the media [1,3,8,9,13,14]. Despite the alerts about the risks of polio resurgence, in 2023, three-dose polio VCRs (considering both OPV and IPV vaccines) were <90% in 89 out of 195 countries, including in Afghanistan and Pakistan, and countries from different WHO regions reported three-dose polio VCRs ≤ 60%, leaving millions of children unprotected [6,7]. Countries with outbreaks caused by cVDPV1 and cVPDV2, like the DRC, have three-dose polio VCR rates that are close to 50% [5].

The surprise was not the emergence of polio outbreaks in countries with low VCRs, but the absence of polio in some of the countries where VCRs are extremely low, such as Ecuador, Haiti, and Vietnam. In 2023, all countries had introduced the first IPV dose (IPV-1), but more than 60% did not achieve VCR ≥ 90% for the IPV-1, and from 85 countries that reported VCR for IPV-2, only 57.6% achieved IPV-2 VCR ≥ 90% [6,7].

In 2021, two cases of AFP caused by cVDPV2 were confirmed in Ukraine (the last on 24 December 2021) [5]. The three-dose polio VCR rate in Ukraine in 2021 was 78%, and it dropped to 69% in 2022 because of the war [6,7]. With the persistence of the conflict, the risk of polio resurgence is remarkably high in this country.

The risk of polio resurgence does not exist only in countries with low VCRs. In 2022, one AFP case caused by cVDPV2 was confirmed in the USA and one, caused by cVDPV3, in Israel. In 2023, Israel also reported one AFP case caused by cVDPV2. These two countries have high VCRs and good infrastructure to investigate and organize catch-up immunization in at-risk areas, impairing the spread of VDPVs [5]. In the UK and Canada, cVDPV2 was detected in 2022 through ES without causing paralysis, and the strains detected in the USA, the UK, and Canada were genetically related. In these three countries, only IPV vaccines were used to carry out catch-up vaccination, with no new AFP case or detection of VDPVs in sewage water in 2023 and 2024. In Israel, despite the use of IPV and OPV to control the emergence of cVDPVs, on 19 July 2024, the WHO alerted of the detection of cVDPV2 in six environmental samples in Gaza collected on 23 June 2024 from two different collection sites in two sub-regions [18], and one AFP case caused by cVDPV2 was confirmed on 25 July 2024 [5]. The three-dose polio routine VCR in the occupied Palestinian territory was 99% in 2022 but has declined to 89% after the onset of the conflict, and there is great concern that it will decrease even more as the war drags on [18].

Both OPV and IPV cannot eliminate the risk of polio resurgence in areas with low VCRs.

### 4.2. Polio Surveillance

The other major challenge for polio eradication is surveillance to detect polio infections. Less than 1% of poliovirus infections result in symptomatic disease.

According to the WHO, the strategies for detecting the poliovirus involve AFP surveillance: active surveillance among at least 1/100,000 individuals younger than 15 years of age in non-endemic countries, and passive surveillance with investigation of notified AFP cases with 80% adequate stool collection, supplemented by the environmental surveillance of sewage samples to detect early indicators of silent transmission in a population (i.e., non-paralytic cases) [3,4].

From 2020 to 2024, the number of AFP cases caused by cVDVPs and polio-compatible cases (those for which it is not possible to confirm or exclude polio) were higher than those caused by WPV1 (Figure 1). Additionally, in 2024, there were 11,127 polio-pending cases as of 24 October 2024 (AFP without lab test confirmation) [2].

From 2019 to 2024, according to the WHO, the incidence rate of non-polio AFP cases ranged from 3.83 to 6.53 per 100,000, but the surveillance for AFP is not uniform, and in many regions and countries the AFP surveillance rates are lower than 1/100,000 [2]. Failures in surveillance of AFP cases, the time necessary to confirm the etiology of AFP reported cases, and difficulties to perform laboratory investigations in cases and contacts are important challenges to the timely identification of VDPV circulation and outbreaks, and thus to the timely implementation of control measures, especially in areas with low VCRs and weak surveillance systems [14,20].

In Latin America, for example, no cVDPV cases have been reported [2], but several countries have identified VAPP cases and VDPVs. In the few countries where ES samples were collected, Sabin strains with mutations associated with neurovirulence have been identified [20,21] and rare cases of iVPDV has also been confirmed [20,21,22]. Mexico was the first MIC to introduce an acellular pertussis combination vaccine containing IPV in 2007, after the detection of 2 VAPP cases in healthy infants, but persisted in using OPV for boosters until 2020. Other MICs adopted IPV in RI (standalone or in combination with other antigens) but persisted in using OPV for boosters or during national immunization campaigns. More recently, a greater number of MICs are using only IPV, like Argentina, Chile, Uruguay, and Mexico.

Independently of the rarity of AFP cases caused by VDPV strains, VAPP cases have been confirmed in different countries, in addition to some iVDPV cases and some ambiguous AFP cases [2,10,20,21,22]. Many countries of this region reported exceptionally low three-dose polio VCRs after the COVID-19 pandemic. In Venezuela, five polio-compatible cases were registered in 2020, and six in 2022. It is impossible to know if they were caused by poliovirus, but considering the extremely low three-dose polio VCR (51%) and the political situation in this country, the probability that some are cVDPVs is high [2].

In regions where the population is seriously under-immunized, and a high number of susceptible children are immunized with OPV, Sabin strains can circulate for a prolonged period, which increases the probability to mutate and, over the course of 12–18 months, to reacquire neurovirulence [1,8,9]. On 27 December 2022, one AFP case was reported in Peru in a non-immunized healthy infant. Only in March 2023, three months thereafter, the genetic characterization of the strain was classified as VDPV type 1, and a regional alert was issued by the PAHO in April (4 months after case detection) [23]. In Peru and Venezuela there is no ES, and in Guatemala, where one VDPV case was reported, a retrospective trial showed that more than 30% of AFP cases were not reported through the AFP surveillance system [24].

Polio cases can be missed by AFP cases surveillance alone, emphasizing the importance of ES. In China, a total of 624 sewage samples were collected from domestic sewage in Guangzhou City, from 2009 to 2021. Poliovirus was identified in 66.67%. After inoculation and replication in three cell lines, 3370 viruses were isolated during a 13-year surveillance period. Among these, 1086 isolates were identified as polioviruses: type 1 PV (21.36%), type 2 PV (29.19%), and type 3 PV (49.48%). Based on VP1 capsid protein genome sequences, 1057 strains were identified as Sabin-like, 21 strains were high-mutant vaccines, and eight strains were classified as VDPVs. The numbers and serotypes of poliovirus isolates in sewage were influenced by the vaccine switch strategy. After 2016, no Sabin-2 was isolated from sewage and type 3 PV isolates increased significantly and became the dominant serotype [25]. The risk for cVDPV3 outbreak reemergence did not disappear from China, where on 25 January 2023 one cVDPV3 was detected through ES [5].

China introduced sIPV in its RI in 2015, and reports an extremely high VCR for three-dose polio vaccination [6,7]. The origin of this cVDPV3 strain is unknown.

The time for the detection of wild polio virus and VDPVs detected in AFP cases or through ES is key to the implementation of immunization campaigns. As an example, in Israel, cVDPV3 strains were detected through ES in 2021, and the AFP case caused by this strain was confirmed in 2022 [5].

Between 2004 and 2019, in a total of 74 outbreaks caused by cVDPVs registered in 24 countries, the median time from seeding until outbreak detection for serotype 1 was 572 days (95% CI 279–2016), in 10 outbreaks; for serotype 2 (in 59 outbreaks), it was 276 days (95% CI 172–765), and for serotype 3 (in 5 outbreaks), was 472 days (95% CI 392–603) [10].

The same problem related to a delay in the identification of polio strains was noted with WPV1. In the recently isolated WPV1 in Malawi, extensive nucleotide changes were detected, indicating that WPV1 had been circulating in southeastern Africa for approximately 2 years before the detection of these strains [12].

Recently, the time for detection of VDPVs seems to have been reduced, as observed in the USA, when one case of AFP was confirmed in 2022, and the genetic analysis of the cVDPV2 isolated from this case was related to strains identified through ES in the UK only a few months before [3]. In Yemen, cVDPV2 was isolated from 227 AFP cases and from 83% (39/47) of environmental samples with an average of 7 months’ delay from sample collection [26]. Regardless of this reduction in detection time, this was insufficient to implement the campaigns of immunization to control cVDPV2 outbreaks in African countries [27]. Additionally, a review of VAPP cases proved the strong correlation with immunodeficiencies and also that clinical manifestations can develop 2 months to 4 years after the last OPV dose [28]. Immunocompromised people can excrete the virus for years and infect close contacts that can also shed the virus, maintaining silent circulation and transmission in the community [28].

The most recent alert from the WHO about the emergence of the cVDPV2 strain in Gaza reported that the genomic sequencing of the six cVDPV2 isolates had close genetic links with each other and are also closely related to the poliovirus variant that was circulating in Egypt during the second half of 2023. In Israel, the last detection of cVDPV2 was in December 2023, and it was estimated that cVDPV2 was introduced in Gaza as early as September 2023. These strains have probably been circulating for 10 months before their detection through ES in Gaza [18]. The isolates have limited divergence from the parental nOPV2 vaccine strain in the VP1 capsid protein coding area (6 to 16 nucleotide substitutions), indicating that surveillance detected emergence early after vaccination [18].

ES is recommended by the WHO to facilitate the identification of VDPVs, contributing to organizing a fast response to outbreaks, but in many outbreak countries, the ES was only introduced after the confirmation of AFP cases. Similarly to VCRs, ES is not uniform. As an example, the DRC, where 75% of cVDPV1 AFP cases confirmed in 2021 and 2022 were reported, had no VDPV1 identified by ES, and Madagascar, where 12% of cVDPV1 were reported, reported more cVDPV1 by ES from 2021 to 2024 [5].

### 4.3. Benefits and Risks Associated with Live and Inactivated Vaccines Recommended by the WHO

The WHO estimates that the large use of oral polio vaccines prevented more than 6.5 million children from developing polio caused by WPV in the last decade [1], but despite the success of the tOPV vaccine in eradicating WPV2 and WPV3 and the huge reduction of countries infected by WPV1, the emergence of VDPVs is now a difficult-to-control public health emergency.

To reduce the risk of polio, in many countries where VCRs are low, the strategy is to start the RI with bOPV followed by IPV, because IPV can boost the serologic and mucosal immunity in infants previously immunized with OPV. Many low-income countries also include a dose of OPV to newborns in an attempt to improve the VCRs. Even though the OPV and nOPV2 vaccines are considered safe for newborns, most immunodeficiencies are not diagnosed in the first months of life, and many times, AFP is the first alert to investigate immunodeficiency [22]. Recently, in China, one 7 month-old immunocompromised child developed AFP and infected his father, who shed the virus without any symptom [29]. In China, the genetic evolution of iVDPV3 showed the recombination of different Sabin strains [30].

The lower seroconversion of OPV observed in many developing countries was considered responsible for AFP cases in children who received five or more doses of OPV. This problem was detected especially in areas where there is a high circulation of other enteroviruses, low hygiene, and malnutrition, exactly those where it was more difficult to eliminate WPV and cVDPVs [8,9]. In addition, when tOPV was substituted for bOPV1,3, there was a substantial reduction in immunity against polio 2 that was even more reduced after the COVID-19 pandemic. The results of a study in 51 African countries showed that type 2 OPV immunity among children under 5 years declined from a median of 87% in January–June 2016 to 14% in January–June 2020 [27]. The probability of cVDPV2 poliomyelitis among children under 5 years was negatively correlated with OPV-induced and IPV-induced immunity. Campaigns with mOPV2 in response to cVDPV2 outbreaks covered only 11% of children under 5 years who are predicted to be at risk within the first 6 months and only 56% within the first 12 months, resulting in a failure to protect against cVDPV2 spreading [27].

In 2013–2014, Israel experienced an outbreak of wild poliovirus type 1 (WPV1), detected through the environmental surveillance of the sewage system. No cases of AFP were reported, and the epidemic subsided after a bivalent oral polio vaccination campaign. However, Brower et al. [31] estimated that 59% (95% CI 9–77%) of susceptible individuals (primarily children under 10 years old) had been infected with WPV1 over a little more than six months, mostly before the vaccination campaign’s onset, and that the vaccination campaign had averted 10% (95% CI 1–24%) of WPV1 infections.

Many mathematical models have estimated the risks of the emergence of new VDPV strains, especially cVDPV2 and cVDPV1 in areas with extremely low VCR [32,33,34]. Recent publications showed that the introduction of at least one dose of IPV in addition to mOPV or bOPVs can contribute substantially to protect against polio [19,35,36,37,38]. Despite the low impact on intestinal immunity, IPV is highly immunogenic in higher-, middle-, and lower-income countries [8,9]. IPV contributes to boost the serologic response to all strains and/or to offer protection in naïve children and children that did not seroconvert after OPV [36,37,38].

IPV is the safest vaccine against polio because it cannot cause VAPP or VDPVs emergence and according to many experts, in polio-free regions, ethical reasons support the exclusive adoption of the IPV vaccination [38,39]. Children immunized with IPV can be infected by WPV or VDPVs without developing symptoms; these strains can multiply in the gastrointestinal tract and the silent circulation will put at risk people with incomplete protection, whether they live in high-, middle-, or low-income countries. The same can occur when OPV is used in areas where OPV has low VCRs or have demonstrated low efficacy because of the interference of other enteroviruses [8,9].

Devaux et al. (2024) affirmed that the intestinal microbiota can contribute to increasing poliovirus infection, replication, recombination, and transmission. There is evidence that the exposure to bacteria, and especially to lipopolysaccharides, is a potent enhancer of poliovirus infectivity. They also suggested that some bacterial strains increase co-infections by different polioviruses, promoting genetic recombination between different viruses. In LICs, these new aspects related to infection can make the use of IPV more favorable [39].

The viral shedding after the first OPV dose is high, peaking 2 weeks after vaccination, and is null after IPV. In areas with low VCRs, the use of OPV has been responsible for VPDV emergence and cVDPVs outbreaks. Inactivated strains cannot replicate and never cause VAPP or give origin to VDPVs.

Connor et al. (2002), in a review of mucosal immunity against poliovirus, showed that intestinal immunity after oral polio vaccines wanes over time to undetectable levels 1 to 4 years after the primary schedule with OPV. It is not observed also in adolescents and adults, probably because of an age-related defect in enteric mucosal immunity to poliovirus. Older children, adolescents, and adults immunized with OPV more than 1 year before can shed live polio strains for more than 2 months upon challenge. This new knowledge can impact the strategies proposed for the final phase of polio eradication [35].

To solve the problems associated with low seroconversion after immunization with Sabin vaccines, new genetically attenuated live vaccines have been developed over the last decade. The nOPV2 was used since 2021 in epidemic countries, and it was expected that it could reduce the risks of VAPP and VDPV emergences. In fact, it was proven that nOPV2 was 80% less likely to seed new-variant polio outbreaks. However, a lower risk for mutation is different from zero risk, and recently it was confirmed that nOPV2 can also cause paralysis [15,16,17]. Additionally, it also induces lower seroconversion and showed low effectiveness in Nigeria and Libera [40,41], as was seen with classic Sabin OPV vaccines [8,9].

In Liberia, a study including 371 children who had received two doses of nOPV2, showed that only 38.3% had antibodies against type 2 poliovirus, demonstrating that nOPV2 can be poorly immunogenic in some developing countries. In this trial, the seroprevalence against polioviruses type 1 (59.6%) and type 3 (53.0%) was also extremely low, despite the large use of bOPV(1,3) [40].

A recent case-control study performed in Nigeria showed that nOPV2 had low effectiveness to protect against cVDPV2, even after two doses. On the contrary, the IPV demonstrated higher effectiveness and efficacy [41]. Despite the large use of nOPV2 in Nigeria, and the reduction of AFP cases caused by cVDPV2 from 2021 to 2024, >40% of cVDPV2 AFP cases confirmed in 2024 (49 of 117) were from this country [19].

A new cause for concern about nOPV2 was the detection of strains derived from this vaccine in countries that never used it. In South Sudan, a new emergence of cVDPV2 strains was detected with whole-genome sequencing (WGS), indicating that the origin of the virus was the novel OPV2, although the vaccine has never been used in the country [12]. In the Russian Federation, VDPV2 strains detected in migrant children from Tajikistan [42] and in Gaza, cVDPV2 detected through ES were both genetically related to nOPV2 vaccine strains used in Egypt [18].

The other relevant point to discuss is the duration of protection offered by OPVs and IPVs. In the past, the circulation of wild poliovirus could provide a natural booster, but large outbreaks involving adolescents and adults demonstrated that it is not possible to guarantee this natural protection [8,9]. Gao et al. (2022), in a meta-analysis about the duration of protection after vaccines, reported only eight papers discussing polio duration of protection post-OPV or -IPV, and the majority were focused on combined vaccines containing IPV. The studies with available prevaccination records suggested that the geometric medium titers and geometric medium concentration of antibodies (GMT/GMC) remained around three times higher than prevaccination levels for all three poliovirus types at 5–6 years postvaccination, but only one article reporting results for 10 years postvaccination was included in this meta-analysis. There were no data about the duration of protection offered by OPV in LICs/MICs in the absence of poliovirus circulation and campaigns with OPV [43]. A study performed in Italy showed a long duration of protection with both OPV and IPV, but Italy is a HIC, where OPV has higher immunogenicity as compared with MICs/LICs [44].

A cross-sectional study for poliovirus seroprevalence performed in Guangdong, China, carried out in 2014 (before the change in the polio vaccine in 2015), including 6339 people, showed that the seropositivities for PV1, PV2, and PV3 were 95.2%, 94.9%, and 88.7%, and the respective GMTs were 82.9, 55.8, and 26.3, respectively. The highest seropositivity and GMT were observed in the 1–9-year-old age group. However, similarly to other seroprevalence studies, lower immunogenicity was observed for PV3 [45]. Another study conducted in Chongqing, China tested 636 people for poliovirus neutralizing antibodies. The overall seroprevalence rates for PV1, PV2, and PV3 were 93.40%, 96.38%, and 91.82%, and GMTs were 61.14, 66.78, and 21.47, respectively. GMTs were negatively associated with age in the geographic districts with poor economic conditions, which will increase the risk of the emergence of VDPVs after a polio vaccine switch. More than one dose of IPV should be introduced into the polio vaccine schedule, and the supplementary immunization of polio should still be annually carried out after the polio vaccine switch, especially among older children and adults [46].

In the DRC, an outbreak was caused by WPV1 in 2010, affecting 16% of adults. It is now one of the countries with the largest number of AFP cases caused by cVDPV1 and cVDPV2. A study assessing poliovirus seroprevalence in 5526 adults aged 15–59 years was performed in this country in 2013–2014. The results showed that 74%, 72%, and 57% of adults were seropositive for neutralizing antibodies for poliovirus types 1, 2, and 3, respectively. These results show that adults can be at risk also for cVDPVs [47]. Two doses of full-dose IPV can elicit more than 90% protection against polio, but HICs adopted three or more doses to ensure long-lasting protection, and all trials that estimated the duration of protection included a preschool IPV dose [9]. IPV can boost the immunity to polio 1 and 3 after bOPV1,3, but two IPV doses can be insufficient to ensure a long duration of protection against polio 2. In countries that use fIPV, despite the good seroconversion after two doses, the antibody titers are lower as compared with full IPV, and there is no study confirming the long-term protection offered by this vaccine [9]. India recently adopted three fIPV doses (the last one at 9 months concomitantly with measles) [3,11].

### 4.4. Plannings for the End of the Game and Post-Eradication Era

In the last decade we learned that we should consider not only the advantages and disadvantages of each type of vaccine, but also other factors, like logistics, acceptance, supply, regulatory challenges involving clinical trials and time for approval, and the industrial capacity to supply the new vaccines to LMICs [19,48,49,50,51,52].

Considering that WPV1 and cVDPVs were not eradicated until the middle of 2024, a new plan for polio eradication will be proposed by the GPEI, and the best strategies for this phase are debated [39,48,49,50,51]. The strategy proposed by the GPEI to eradicate polio assumes that the route of transmission of WPV and cVDPVs is feco-oral, but this hypothesis remains controversial, and recent evidence based on the transmission of cVDPVs showed that there is a higher possibility that the route is mostly respiratory rather than only feco-oral, which is one more factor to support the exclusive use of IPV [51].

There is no doubt that only IPV can eradicate both WPV and cVDPVs [1]. Combination vaccines containing IPV and three (DPT) or more antigens (Hib, HepB) have been extensively used in HICs, and in some MICs, showing an excellent safety profile, improving timely vaccination, and leading to higher VCRs for many serious diseases. China and Japan adopted an IPV based on Sabin strains, approved since 2012, and a few LMICs adopted an fIPV, which cannot be combined with other antigens. The biggest challenge persists in the poorest countries, exactly where VCRs are low, OPVs induce lower seroconversion, intestinal infections are frequent, and there is recurrent civil unrest, conflicts, and natural disasters that have dramatic impacts on public services [52,53,54,55]. The lack of access to medical services, combined with the destruction of the already weak infrastructure (water, sewage, distribution of food and medicines), have been responsible for population displacements. The great number of migrants from regions where oral vaccines are still used and the travelers to these countries, including military, healthcare professionals, and volunteers, can contribute to the importation of polioviruses into areas considered polio-free [8,9,12,14,17,18,19,39,43,44,49,50,51,52,53,54].

The WHO considers that there is a sufficient supply of IPV (including fIPV and Sabin IPV), but the estimation was made for only two doses. If a transition to the exclusive use of an IPV or at least more than two doses of an IPV is expected, close coordination with manufacturers should be engaged to account for the time required to ramp up production and deliver sufficient capacity. Additionally, it is fundamental to consider that many people older than 5 years were not protected, and catching up will be necessary for non-immunized children, adolescents, and adults, increasing the need for an IPV supply [56].

There are some new vaccines in development proposed to finish polio eradication. Among the candidates, those produced with live strains (including nOPV1, nOPV3, and nOPV1,3) can pose the same challenges as recently discovered with nOPV2 (transmission, a low but confirmed risk of mutation, and the possibility to cause paralysis). The S19-IPV vaccine, now on phase I of clinical development, is formulated with the S19 virus—a hyper-attenuated, genetically stabilized strain that cannot replicate at human body temperature and has been highly restricted in its potential to revert to a neurovirulent form during the manufacturing process. It is impossible to know if this strain could recombine with other Sabin strains or enterovirus in the case of an escape from manufacturing sites [50].

The VLP vaccine is also a promise, not a reality. There are substantial regulatory challenges for its approval. It will not be possible to demonstrate its efficacy/effectiveness against WPV and VDPVs. It will be difficult to prove if it will induce intestinal immunity and if it can be combined with other antigens. The clinical development plan to have pentavalent or hexavalent combinations with new vaccines requires time and funding that could be better invested in combination vaccines containing IPV that are already available and have confirmed efficacy/effectiveness against polio and other diseases, like diphtheria, pertussis, tetanus, Hib, and Hepatitis B, for which the burden and case fatality rates are higher in the poorest countries.

## 5. Conclusions

The lessons learned from recent years show that the plannings to substitute tOPV for bOPV, IPV, and nOPVs did not precisely consider the low VCRs and the necessity of a large global supply of vaccines, resulting in a substantial number of outbreaks.

VCRs are not uniform, and the difficulties in implementing environmental surveillance lead to the frequent underestimation of the occurrence of gaps in immunity, as well as of the silent circulation of polioviruses, before and after global polio eradication.

The recommendation for two doses of IPV makes the routine immunization more complex, but IPV can be combined with other antigens to simplify vaccination schedules.

The duration of protection offered by polio vaccines is not known, but considering the evidence of waning antibody titers, it is considered that more than three IPV doses are necessary to guarantee a long duration of protection.

Despite the success of oral vaccines and the hopes that nOPV2 will control cVDPV2 outbreaks, the risks of VAPP and VDPV emergences cannot be acceptable anymore for ethical reasons and social equity.

All countries should make efforts to achieve high and uniform VCRs to minimize the risk of a poliomyelitis outbreak and improve the surveillance of AFP and ES. It will also be necessary to guarantee a sufficient supply and have plans to identify VDPVs and offer a fast response to outbreaks.

Finally, it is key to remember that polio can affect people of all ages, and besides guaranteeing high VCRs in primary series, it is fundamental to guarantee that children, adolescents, and adults (especially healthcare professionals and travelers to risk areas) who were unvaccinated or incompletely vaccinated against polio can also be immunized with IPV.

## Figures and Tables

**Figure 1 vaccines-12-01323-f001:**
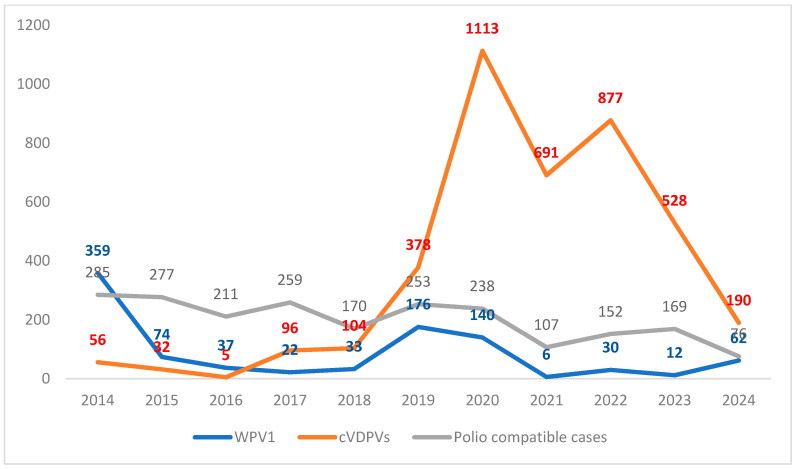
AFP cases caused by WPV, cVDPVs (1,2,3), and polio-compatible cases notified to the WHO from 2014 until 2024 [2]. * Data as of 24 October 2024 [2].

**Figure 2 vaccines-12-01323-f002:**
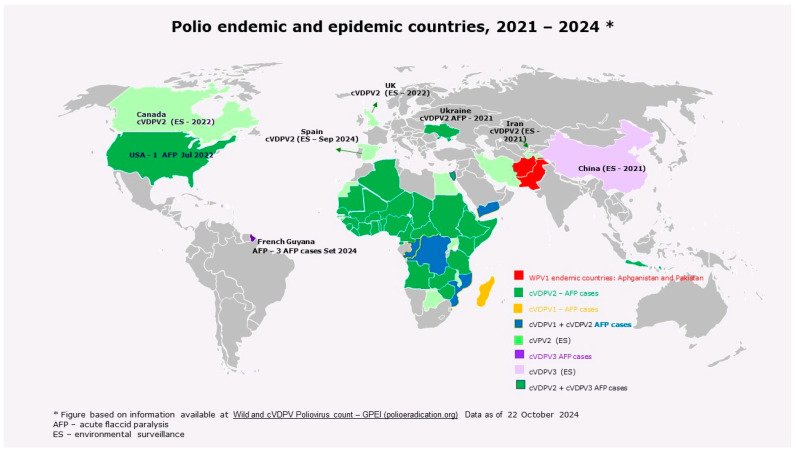
WPV1 polio endemic countries and cVDPVs epidemic countries in the last years (2021 to 2024 as of 22 October 2024).

**Table 1 vaccines-12-01323-t001:** Number of WPV1 and cVPDVs APF cases from 2020 to 2024 [4,5].

AFP	2020 Cases (No. Countries)	2021 Cases (No. Countries)	2022 Cases (No. Countries)	2023 Cases (No. Countries)	2024 Cases (No. Countries) *
WPV1	140 (2)	6 (3)	30 (3)	12 (2)	62 (2)
cVDPV1	35 (4)	14 (2)	192 (5)	134 (3)	8 (2)
cVDPV2	1082 (24)	629 (22)	367 (20)	383 (23)	182 (16)
cVDPV3	0 (0)	0 (0)	1 (1)	0 (0)	0 (0)

* Partial data. Cases reported to the WHO as of 22 October 2024.

## Data Availability

All data were based in the references mentioned in the paper.

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
