# Peer review of "Polio Epidemiology: Strategies and Challenges for Polio Eradication Post the COVID-19 Pandemic"

_vaccines, 2024, doi:10.3390/vaccines12121323_

Round 1
Reviewer 1 Report
Comments and Suggestions for Authors
The authors provide a detailed overview over the Global Polio Eradication Initiative and the successful reduction of wild poliovirus (WPV) cases by over 99.9%, with only 40 cases reported in 2024. Despite the eradication of WPV types 2 and 3, outbreaks of vaccine-derived polioviruses (cVDPVs) have emerged, particularly in regions with low vaccination coverage exacerbated by the COVID-19 pandemic. The review aims to examine the current state of polio epidemiology post-pandemic and the strategies and challenges in achieving global polio eradication. The manuscript is potentially interesting for the scientific community; however, a few minor issues need to be addressed:
1. Page 1, line 21: please define "VCR"
2. Page 2, line 32: "PV" is missing
3. Page 2, line 48: vaccine-derived polioviruses
4. Page 2, line 57: At the onset of
5. Page 2, line 68: "combo" should read "combination"
6. Page 3, line 140: line feed should be removed
7. Page 4, line 158: (Salk strain) should be removed
8. Page 4, line 174: "sufficient protect" should read "sufficient to protect"
9. Page 4, line 176: " resulted in higher" should read " resulted in a higher"
10. Page 5, lines 242-244: It would be beneficial for the reader to have more details about the specific cases. For the US case, what was the VCR in the affected county and surrounding counties? What were the details for the case from Israel?
11. Page 6, line 270: are15,259 >space is missing
12. Page 9, line 408: VDPVs
13. Page 10, line462: Please define the abbreviations GMT/GMC
14. Page 10, line 475: "people poliovirus" should read "people for poliovirus"
15. Page 10, line 499: "that it we" should read "that we"
16. Page 12, line 572: Please define the abbreviation HCP
17. In general, the word "poliovirus" and the abbreviation "PV" are inconsistently used throughout the text. Please correct.
Author Response
Thanks for your valuable suggestions.
We included it in the attached new version, where we also updated the last information about polio for this week (The number of WPV1 now is 62, when we sent the original draft last 27 Sep was 40)
We also included a figure as suggested by another colleague
- Page 1, line 21: please define "VCR" – vaccine coverage rate (VCR)
- Page 2, line 32: "PV" is missing - included
- Page 2, line 48: vaccine-derived polioviruses – already abbreviated
- Page 2, line 57: At the onset of - corrected
- Page 2, line 68: "combo" should read "combination" - corrected
- Page 3, line 140: line feed should be removed - removed
- Page 4, line 158: (Salk strain) should be removed - removed
- Page 4, line 174: "sufficient protect" should read "sufficient to protect" -corrected
- Page 4, line 176: " resulted in higher" should read " resulted in a higher" - corrected
- Page 5, lines 242-244: It would be beneficial for the reader to have more details about the specific cases. For the US case, what was the VCR in the affected county and surrounding
- counties? What were the details for the case from Israel? – can add as a note? It will be necessary another reference
- Page 6, line 270: are15,259 >space is missing OK
- Page 9, line 408: VDPVs - OK
- Page 10, line462: Please define the abbreviations GMT/GMC - included
- Page 10, line 475: "people poliovirus" should read "people for poliovirus" - OK
- Page 10, line 499: "that it we" should read "that we" -OK
- Page 12, line 572: Please define the abbreviation HCP – OK health care professionals
- In general, the word "poliovirus" and the abbreviation "PV" are inconsistently used throughout the text. Please correct. - corrected

Reviewer 2 Report
Comments and Suggestions for Authors
This review focuses on an interesting topic and provides a comprehensive description of the polio epidemiology post-COVID-19 pandemic. Illustrations are strongly suggested for this manuscript, which will make the review easy to understand and follow. The language of this manuscript requires substantial revision. Citations should be put in the proper place in the sentences.
For example, in the Abstract part:
Line 19: should be “the covid-19”,
Line 20: should be “the WHO estimates…”
Line 21: what is “VCR”
Line 22: should be “pose a risk…”
…
…
…
Comments on the Quality of English LanguageThe language of this manuscript requires substantial revision.
For example, in the Abstract part:
Line 19: should be “the covid-19”,
Line 20: should be “the WHO estimates…”
Line 21: what is “VCR”
Line 22: should be “pose a risk…”
…
…
…
Author Response
Thanks for your valuable suggestions. We corrected in the paper, revised the English and also added a figure to make more clear. We also updated the information available at WHO sites (WPV1 now is 62 , when we sent the first version last 27 Sep was 40!)
Reviewer 2 Polio
This review focuses on an interesting topic and provides a comprehensive description of the polio epidemiology post-COVID-19 pandemic. Illustrations are strongly suggested for this manuscript, which will make the review easy to understand and follow. The language of this manuscript requires substantial revision. Citations should be put in the proper place in the sentences.
For example, in the Abstract part:
Line 19: should be “the covid-19”, - OK
Line 20: should be “the WHO estimates…” - OK
Line 21: what is “VCR” - vaccine coverage rate (inserted in the new version)
Line 22: should be “pose a risk…” - OK
Comments on the Quality of English Language
The language of this manuscript requires substantial revision.
After review the language, we modify the abstract, and made de corrections suggested. We also included a figure to make more clear the evolution of WPV and cVPVs, and updated data according the last information available at WHO sites.

Reviewer 3 Report
Comments and Suggestions for Authors
This paper provides a comprehensive compendium of information on polio vaccination options, rates, and ongoing transmission of poliovirus globally. It shows a worrisome trend, that we are not rid of this virus worldwide.
Your discussion of how to approach eradication going forward, is thorough and well researched. It would be helpful if you could contrast your thoughts with the WHO or GPEI plans.
You present a large amount of detailed information, on cases by type and country. It is difficult to follow or to know the relevance of all the details. Would it be possible to present this information in chart form? E.g. perhaps a time line by country by type of virus; and also a map with last reported case by country by type and number. Tables 1 and 2 are very helpful, but it is difficult to keep track of the large volume of additional information presented in the text.
A list of acronyms and abbreviations would also be helpful. You explain most in the introduction but by the end of the paper a reader may not remember what an acronym or abbreviation stood for. Also do make sure you spell out each with first use. For example you use VCR in the abstract but do not explain that until the introduction.
There are some word choices that could be improved. e.g. page 4, Line 177-178, and also page 8 line 370: "tentative". I think you mean "attempt" or "Strategy".
A second word choice issue is on page 5. lines 204 and 206, "WHO informed".. I think you mean WHO reported...
Page 5 line 228: You are noting inadequate vaccine coverage rate, and report on the number of countries with a rate <=90%. Earlier you say that adequate rate is >=90%. 90% cannot be in both places; I think you should report on the number of countries with a vaccine coverage rate <90%.
Page 6 line 263: I am not sure what is the meaning of "active surveillance among at least 1/100,000 individuals...". Please explain or rephrase.
Page 7 line 303: "Trial".. Do you mean study?
Page 7 line 326: "seeding": I think this means when the virus was introduced into the community, but please put in the correct definition.
Comments on the Quality of English LanguageExcellent English, some word choices could be improved as noted. Also there are some places where tense of verb does not match subject.
Author Response
Comments and Suggestions for Authors
This paper provides a comprehensive compendium of information on polio vaccination options, rates, and ongoing transmission of poliovirus globally. It shows a worrisome trend, that we are not rid of this virus worldwide.
Your discussion of how to approach eradication going forward, is thorough and well researched. It would be helpful if you could contrast your thoughts with the WHO or GPEI plans.
You present a large amount of detailed information, on cases by type and country. It is difficult to follow or to know the relevance of all the details. Would it be possible to present this information in chart form? E.g. perhaps a time line by country by type of virus; and also a map with last reported case by country by type and number. Tables 1 and 2 are very helpful, but it is difficult to keep track of the large volume of additional information presented in the text.
1. First of all, thanks a lot for your valuable comments and suggestions. It was very challenging to organize so huge quantity of new information, and many interesting references we need to cut, to summarize the most relevant ones.
After discussing about the use or not of figures, we decided to substitute the table 2 by Figure 1, but we did not include a map, because all relevant information (country by country) can be find in the reference sites, that we updated in this new version as of today. Data are changing every week: last 27 Sep (first draft) there were 40 WPV1 AFP cases, but now the number is 62!
2. A list of acronyms and abbreviations would also be helpful. You explain most in the introduction but by the end of the paper a reader may not remember what an acronym or abbreviation stood for. Also do make sure you spell out each with first use. For example you use VCR in the abstract but do not explain that until the introduction.
We reviewed the acronyms and abbreviations, and we think that now they are OK. About the list we will consult the editor if we can insert it.
3. There are some word choices that could be improved. e.g. page 4, Line 177-178, and also page 8 line 370: "tentative". I think you mean "attempt" or "Strategy".
A second word choice issue is on page 5. lines 204 and 206, "WHO informed".. I think you mean WHO reported...
Page 5 line 228: You are noting inadequate vaccine coverage rate, and report on the number of countries with a rate <=90%. Earlier you say that adequate rate is >=90%. 90% cannot be in both places; I think you should report on the number of countries with a vaccine coverage rate <90%. OK
Page 6 line 263: I am not sure what is the meaning of "active surveillance among at least 1/100,000 individuals...". Please explain or rephrase. This is the WHO recommendation to identify the cases of acute flaccid paralysis. As it can be caused by other enterovirus, even in non-endemic countries, the minimun report should be 1 by 100,000 individuals. As we discussed, the medium of AFP cases reported globally is 5 by 100,000, but it varies a lot, and many countries reported less than 1/100,000.
Page 7 line 303: "Trial".. Do you mean study? Yes.
Page 7 line 326: "seeding": I think this means when the virus was introduced into the community, but please put in the correct definition.
Virus shedding is the elimination of the virus by people immunized with live vaccines (OPV) that does not occur with inactivated ones (IPV)
4. Comments on the Quality of English Language
Excellent English, some word choices could be improved as noted. Also there are some places where tense of verb does not match subject. - we made a review and included also the suggestions of other reviewers (the number of pages is not the same)
Round 2
Reviewer 3 Report
Comments and Suggestions for Authors
There is a great deal of information in text. A table and map collating the majority of information on outbreaks and isolates would be useful, and the text can be shortened.
Author Response
Comments: There is a great deal of information in text. A table and map collating the majority of information on outbreaks and isolates would be useful, and the text can be shortened.
Response: We constructed a map to show polio endemic and epidemic countries infected by different cVDPVs added in Supplementary material. However, we believe that it´s not possible to cut the text because it contains relevant information countries that are used to discuss the strategies.
In the last 4 years, countries have been infected and reinfected by cVDPVs, and many also have detections of polio through environmental surveillance without APF cases detection, including countries with high vaccine coverage rates (Canada, US, UK, Israel, China, and recently Spain). The situation is very dynamic and changes a lot, for example, the number of WPV1 in the first draft was 40, in the version updated is 62 (until October 24) and today 68!